# Interelastomer Reactions Occurring during the Cross-Linking of Hydrogenated Acrylonitrile-Butadiene (HNBR) and Chloroprene (CR) Rubbers Blends in the Presence of Silver(I) Oxide (Ag_2_O) and Mechanical Properties of Cured Products

**DOI:** 10.3390/ma16134573

**Published:** 2023-06-25

**Authors:** Aleksandra Smejda-Krzewicka, Konrad Mrozowski, Krzysztof Strzelec

**Affiliations:** Institute of Polymer and Dye Technology, Lodz University of Technology, Stefanowskiego Street 16, 90-537 Lodz, Poland; 225373@edu.p.lodz.pl (K.M.); krzysztof.strzelec@p.lodz.pl (K.S.)

**Keywords:** silver(I) oxide, chloroprene rubber, hydrogenated acrylonitrile-butadiene rubber, cross-linking mechanism, morphology, thermal analysis

## Abstract

The purpose of this paper was to examine the possibility of producing new blends of hydrogenated acrylonitrile-butadiene and chloroprene rubbers (HNBR/CR) unconventionally cross-linked with silver(I) oxide (Ag_2_O), and to investigate the physicomechanical properties of the obtained materials. From the obtained results, it can be concluded that HNBR/CR composites were effectively cured with Ag_2_O, which led to interelastomer reactions, and the degree of binding of HNBR with CR was in the range of 14–59%. The rheometric and equilibrium swelling studies revealed that the cross-linking progress depended on the weight proportion of both elastomers, and the degree of cross-linking was greater with more content of chloroprene rubber in the tested blends. Interelastomer reactions occurring between HNBR and CR improved the homogeneity and miscibility of the tested compositions, which was confirmed by differential scanning calorimetry (DSC) and scanning electron microscopy (SEM) analyses. The tensile strength and hardness of the obtained HNBR/CR/Ag_2_O vulcanizates proportionally increased with the content of CR, while the tear strength showed an inverse relationship. The obtained new, unconventional materials were characterized by significant resistance to thermo-oxidative factors, which was confirmed by the high aging factor.

## 1. Introduction

Hydrogenated acrylonitrile-butadiene rubber (HNBR) is well-known for its resistance to chemical and thermal degradation and can be formulated with varying acrylonitrile contents and degrees of hydrogenation. HNBR finds many applications that combine relatively high temperatures with exposure to petroleum products, such as automotive engine bay applications and sealing and piping equipment for the oil industry [1,2,3]. In both of these application areas, high pressures and temperatures approaching 200 °C create an aggressive environment for polymers, with peak engine bay temperatures predicted to increase in the future, and deeper, more challenging subsea oilfield locations are being targeted for future exploitation [1,2].

In the case of HNBR, the type of cross-linking agent used depends on the degree of unsaturation of the elastomer. Fully hydrogenated HNBR (<1% of the initial number of double bonds), due to its saturated nature, can only be cross-linked with organic peroxides (usually in the presence of coagents) or by ionizing radiation. On the other hand, the cross-linking of partially hydrogenated HNBR (2–5% of the initial number of double bonds) can be classically carried out with sulfur in the presence of ultra- or semi-ultra-accelerators or with tetraalkylthiuram disulfides as sulfur donors [4,5,6]. Compared to peroxide-cured HNBR, sulfur-cured HNBR products have higher tear and tensile strengths, but, unfortunately, they exhibit worse resistance to thermal and ozone aging and higher permanent deformation [4,5]. Currently, the use of high-energy ionizing radiation as a method of HNBR cross-linking is of increasing importance in the industry. Radiation vulcanization can be carried out at ambient temperature without chemical initiators, while the conditions of the process (radiation dose, its intensity, and the process environment) are strictly controlled. In this method, there is zero emission of by-products, in contrast to conventional methods [4].

The use of chloroprene rubber (CR) in composites with HNBR can improve mechanical properties as well as flame resistance. Chloroprene rubber is a speciality elastomer, characterized by good mechanical properties, good adhesion, resistance to technical media, crystallization ability, and thermo-cross-linking ability. An increased flame resistance is the most important parameter characterizing CR. Chloroprene rubber is conventionally cross-linked with zinc oxide (5 phr of ZnO) in the presence of magnesium oxide (4 phr of MgO) and ethylene thiourea (0.3–1 phr of ETU) as an accelerator [7,8,9,10,11]. However, CR can also be cross-linked with other substances, for example, with phenol-formaldehyde resin, sulfur, and its donors [12,13] or other metal oxides.

Metal oxides are inorganic compounds widely used in polymer processing technology. They can be used as cross-linking agents, activators, and fillers as well as desiccants or pigments. Due to their characteristics, metal oxides usually simultaneously fulfill several functions, e.g., a filler and a colorant. In elastomer technology, metal oxides are used to cross-link materials containing atoms belonging to the halogen groups in their main chains. The cross-linking of rubber containing halogen atoms occurs as a result of the reaction of this atom with zinc oxide, resulting in the formation of a Lewis acid (e.g., zinc chloride) [12,14,15], whereas magnesium oxide is an acceptor of the formed hydrogen chloride [15]. Iron(III) and iron(II,III) oxides (Fe_2_O_3_, Fe_3_O_4_) or copper(I) oxide (Cu_2_O) are examples of metal oxides used to cross-link elastomers [16,17,18,19]. An unknown metal oxide used for non-standard cross-linking is silver(I) oxide (Ag_2_O). This oxide occurs as a black powder or as a black-brown oxide with a cubic crystalline structure. Due to its properties, silver(I) oxide is used as an antimicrobial agent. This is because Ag_2_O can inactivate many bacterial enzymes and bind to nucleic acid [20,21,22,23,24]. Therefore, in the textile industry, this oxide has been used as an antibacterial coating for textiles. For polymers, it is mainly used as a reinforcing filler in resins, and it increases hardness and compressive strength and reduces abrasion [25,26,27].

The study aims to unconventionally cross-link HNBR/CR blends using silver(I) oxide and to analyze the properties of the resulting vulcanizates. This research follows EU directives to reduce the use of ZnO because of its harmful effects on the water environment, including organisms. Silver(I) oxide has been used as a cross-linking agent for chloroprene rubber, and the vulcanization gives satisfying results, as confirmed by an earlier publication [11]. However, so far there is no information in the literature on the use of Ag_2_O for cross-linking elastomeric compositions containing chloroprene rubber and hydrogenated acrylonitrile-butadiene rubber.

## 2. Experimental Part

### 2.1. Materials

In this study, hydrogenated acrylonitrile-butadiene rubber (HNBR, Therban 3446) with bound acrylonitrile content: 34% mol., degree of hydrogenation: 94.5% mol., and double bond content [>C=C<]_trans_: 0.49 mol/kg of rubber was provided by Lanxess GmbH (Cologne, Germany). Chloroprene rubber (CR, Baypren 216) with double bond content [>C=C<]_trans_: approx. 80% mol. was produced by Lanxess GmbH (Germany). Silver(I) oxide (Ag_2_O), brand no: AB202471, characterized by a purity of 99% and a density of 7.143 g/mL, was produced by abcr GmbH (Karlsruhe, Germany). Stearic acid (SA) with a density of 0.85 g/mL was produced by Chemical Worldwide Business Sp. z o. o. (Słupca, Poland).

The following solvents were also used: diethyl ether (Chempur, Piekary Slaskie, Poland, solvent with a density of 0.71 g/mL), toluene (POCh S.A., Gliwice, Poland, solvent with a density of 0.87 g/mL), n-heptane (POCh S.A., Gliwice, Poland, solvent with a density of 0.68 g/mL), acetone (Chempur, Piekary Slaskie, Poland, solvent with density of 0.79 g/mL).

### 2.2. Preparation of Elastomeric Composites

HNBR/CR blends were prepared using a Krupp-Gruson laboratory two-roll mill (Laborwalzwerk 200 × 450, Krupp-Gruson, Magdeburg-Buckau, Germany) with a roll diameter of 200 mm and a length of 450 mm. The temperature of the rolls was 20–25 °C, while the speed of the front roll was 20 rpm, and the roll’s friction was 1:1.25. At the beginning, each of the rubbers used was plasticized for 1 min. Then, both rubbers were mixed, and stearic acid was added into HNBR/CR matrix (within 2 min). Finally, silver(I) oxide was incorporated. The blend was mixed for 2 min, and then it was stored in foils at room temperature. The prepared blends were vulcanized using an electrically heated hydraulic press. The vulcanization parameters were as follows: temperature—160 °C, pressure—200 bars, and time—15 min. The preparation of the samples consisted of cutting the prepared blends with a weight depending on the shape of the vulcanization mold. A polyethylene terephthalate (PET) foil was used to make it easier to remove the molded parts from the molds, which were removed after the process. The samples prepared as described above were left for 24 h at room temperature for conditioning.

### 2.3. Characterization of Elastomeric Composites

#### 2.3.1. Cross-Linking Characteristics

Vulcametric measurements were determined by Alpha Technologies MDR 2000 rotorless rheometer (MDR 2000, Alpha Technologies, Hudson, OH, USA), heated to 160 °C. The oscillation frequency was 1.67 Hz. The test lasted 30 min and was performed according to ASTM D5289-17 standard [28]. The test consisted of registering the torque as a function of time during the heating at a constant temperature. The torque value depends on the stiffness of the rubber compound and changes as the cross-linking process progresses. Based on the vulcametric curves, the optimal cross-linking time (t_90_—the time at which the torque reaches 90% of the increase), scorch time (t_02_), minimal torque (T_min_), and torque increment after a specified heating time (ΔT_x_) were determined. The cure rate index (CRI), a measure of the cross-linking rate, was calculated by Formula (1), and the torque increment after a given time of heating was calculated by Formula (2):
(1)
$$CRI=\frac{100}{t_{90}-t_{02}}$$


(2)
$${∆T}_{x}=T_{x}-T_{min}$$


The determination of equilibrium swelling was performed. Samples were cut from prepared vulcanizates in four different shapes. Each of them weighed 25 to 50 mg, with an accuracy of 0.1 mg. The samples were then placed with solvents, toluene or n-heptane, in weighing vessels. The prepared samples were placed in a thermostatic chamber for 72 h at 25 °C, which was then bathed with diethyl ether, dried on filter paper, and weighed again. Next, the samples were dried in a dryer at a temperature of 50 °C to a constant weight and were reweighed.

#### 2.3.2. Resistance to Swelling

The equilibrium volume swelling in toluene or n-heptane (Q_v_) was calculated by Formula (3):
(3)
$$Q_{v}=\frac{m_{s}-m_{d}}{m_{d}^{*}}\times \frac{d_{v}}{d_{s}}$$

where m_s_ is the swollen sample weight (mg), m_d_ is the dry sample weight (mg), d_v_ is the vulcanizate density (g/mL), d_s_ is the solvent density (g/mL), and 
$m_{d}^{*}$
 is the reduced sample weight determined by Formula (4):
(4)
$$m_{d}^{*}=m_{d}-m_{0}\times \frac{m_{m}}{m_{t}}$$

where m_0_ is the initial sample weight (mg), m_m_ is the mineral substances content in the compound (mg), and m_t_ is the total weight of the compound (mg).

The content of the eluted fraction in toluene (W_Q_) of vulcanizates was calculated by Formula (5):
(5)
$$W_{Q}=\frac{m_{0}-m_{d}^{*}}{m_{0}}$$


The theoretical content of the eluted fraction eluted (
$W_{Q}^{th}$
) of vulcanizates was determined by Formula (6):
(6)
$$W_{Q}^{th}=W_{QCR}^{th}\times U_{CR}+W_{QHNBR}^{th}\times U_{HNBR}$$

where 
$W_{QCR}^{th}$
 is the theoretical content of the eluted fraction of CR, 
$W_{QHNBR}^{th}$
 is the theoretical content of the eluted fraction of HNBR, U_CR_ is the weight fraction of CR in the compositions, and U_HNBR_ is the weight fraction of HNBR in the compositions.

The degree of binding of hydrogenated acrylonitrile-butadiene rubber into the interelastomer network with chloroprene rubber (HNBR_b_) was calculated by Formula (7):
(7)
$$HN{BR}_{b}={HNBR}_{0}\times \frac{W_{Q}^{th}-W_{Q}}{W_{Q}^{th}}$$

where HNBR_0_ is the weight fraction of hydrogenated acrylonitrile-butadiene rubber in the HNBR/CR blend.

The volume fraction of rubber in swollen materials (V_r_) was determined by Formula (8):
(8)
$$V_{r}=\frac{1}{1+Q_{v}}$$


The degree of cross-linking (α_c_) was determined by Formula (9):
(9)
$$\alpha_{c}=\frac{1}{Q_{v}}$$


#### 2.3.3. Analysis of Infrared Spectra (ATR-FTIR)

HNBR/CR composites were characterized by Attenuated Total Reflectance Fourier-Transform Infrared (ATR-FTIR) spectroscopy (Nicolet 6700 FT-IR Spectrometer, Thermo Fisher Scientific, Waltham, MA, USA). The spectra were recorded in the wavelength range of 4000–400 cm^−1^ using a Thermo Scientific Nicolet 6700 FTIR spectrometer (Thermo Fisher Scientific, Waltham, MA, USA). Samples for infrared studies were prepared from rubber blends before and after their vulcanization.

#### 2.3.4. Differential Scanning Calorimetry (DSC)

The thermal changes and cross-linking temperature range of the elastomeric blends were investigated using a DSC1 from Mettler Toledo (Mettler-Toledo, Columbus, OH, USA). The thermal properties were measured over a temperature range of −120 to 250 °C, at a heating rate of 10 °C/min, and using liquid nitrogen as a coolant. Before measurement, the DSC analyzer was calibrated using two calibrators: indium and n-octane.

#### 2.3.5. Scanning Electron Microscopy (SEM)

The surface morphology of vulcanizates was evaluated using a scanning electron microscope (SEM): Hitachi Tabletop Microscope TM-1000 (Tokyo, Japan). The preparation of samples for measurement consisted of placing a double-sided self-adhesive foil on special tables and gluing the testing sample to it. Then, a gold layer was applied to the prepared sample using the Cressington Sputter coater 108 auto vacuum sputtering machine (Redding, CA, USA) at a pressure greater than 40 mbar, for 60 s. The sample prepared in this way was placed in a scanning electron microscope chamber and the measurement was performed. The distribution of elements in selected micro-areas and their mapping were made using scanning electron microscopy by Hitachi S-4700 (Tokyo, Japan) with the ThermoScientific energy-dispersive spectrometer (EDS) microanalysis adapter.

#### 2.3.6. Determination of Mechanical Properties

The tensile properties were measured according to the PN-ISO 37:2017 [29] standard using a ZwickRoell machine (model 1435, Ulm, Germany) connected with the appropriate computer software. In the study, samples in the shape of B-type paddles with a measuring section width of 4 mm were used. The scope of the properties tests included: stress at 100%, 200%, and 300% of elongation (S_e100_, S_e200_, and S_e300_, respectively); tensile strength (TS_b_); elongation at break (E_b_).

The tear strength (T_s_) was measured using the A method by PN-ISO 34-1:2010 using a ZwickRoell machine (model 1435, Ulm, Germany) connected with appropriate computer software. Rectangular-shaped samples with dimensions of 100 mm × 15 mm and a cut of 40 mm were used for the tests.

The hardness (HA) was measured with a ZwickRoell (Ulm, Germany) hardness tester by ISO 48-4:2018. The samples for this test were prepared in the shape of cylinders in a specially prepared form. The measurement results were determined on the Shore A scale.

#### 2.3.7. Thermo-Oxidative Aging

The thermo-oxidative aging was tested using the Geer method. For this purpose, the samples to be measured were made in the same way as for the tensile strength test. Five B-type paddles with a measuring section width of 4 mm were cut from each vulcanizate. The prepared samples were placed in a forced-air thermostat at 70 °C. After 7 days, the samples were removed to investigate changes in their strength properties using a ZwickRoell machine (model 1435, Ulm, Germany). Based on the obtained TS_b_ and E_b_ parameters, the aging index (AF) was calculated for the individual vulcanized products by Formula (10):
(10)
$$AF=\frac{{{TS}_{b}}^{*}\times {E_{b}}^{*}}{{TS}_{b}\times E_{b}}$$

where TS_b_ is the tensile strength, E_b_ is the elongation at break, TS_b_^*^ is the tensile strength after thermo-oxidative aging, and E_b_^*^ is the elongation at break after thermo-oxidative aging.

## 3. Results and Discussion

Elastomeric blends are of great importance in the processing of polymers because they are characterized by high performance parameters, durability, well-balanced mechanical properties, and a relatively low cost [30,31]. Therefore, rubbers blends have gained considerable attention in various applications in all fields of industry and technology [32]. However, in order to produce rubber goods with the properties of construction materials from elastomeric blends, they must be properly vulcanized.

To investigate the ability of blends of chloroprene rubber and hydrogenated acrylonitrile-butadiene rubber to cross-link with silver(I) oxide, compositions with different weight ratios of HNBR and CR (HNBR/CR = 100/0, 80/20, 75/25, 50/50, 60/40, 80/20, and 0/100) were prepared (Table 1). For each blend, the same amount of silver(I) oxide (2.5 phr) was used to cross-link. For comparison, samples of pure chloroprene rubber and pure hydrogenated acrylonitrile-butadiene rubber were also prepared. The purpose of using silver(I) oxide was to obtain vulcanizates with better physicomechanical properties in comparison to conventionally cross-linked materials and to provide an alternative to zinc oxide.

### 3.1. Cross-Linking Mechanism of HNBR/CR Composites in the Presence of Silver(I) Oxide

Based on the tests carried out, it was found that during the heating of a blend containing chloroprene rubber and hydrogenated acrylonitrile-butadiene rubber in the presence of silver(I) oxide, interelastomer bonds formed between the tested elastomers. Before presenting the scheme of the interelastomer connection of CR with HNBR, it is important to recall which units in the tested rubbers are important for cross-linking. In the main chain of HNBR there are ethylene units, 1,2-butylene units, acrylonitrile units, and 1,4-*trans*-butadiene units, which are most likely involved in cross-linking (Figure 1).

In contrast, chloroprene rubber is obtained at an elevated temperature as a result of free radical emulsion polymerization, it is characterized by high structure regularity, and its macromolecules mainly consist of 1,4-*trans*-chloroprene units and 1,4-*cis*-chloroprene units (Figure 2). In addition, 1,2- or 3,4-chloroprene units are also present in the main chain [33,34]. The influence of the electronegative chlorine atom bound to the carbon atom at the double bond is so strong on the double bond that 1,2-chloroprene units (when the chlorine atom is in the allyl position and exhibits high reactivity) participate in CR cross-linking [35]. It is worth noting that the 1,2-chloroprene units are isomerized to the form shown in Figure 3 [35].

In HNBR/CR blends, the double >C=C< bonds (1,4-*trans*) of hydrogenated acrylonitrile-butadiene rubber and units with the vinyl addition of chloroprene rubber participate in the cross-linking process of HNBR/CR composites. The cross-linking of HNBR/CR blends in the presence of silver(I) oxide begins with the dehydrohalogenation of chloroprene rubber with the release of hydrogen chloride (HCl). Hydrogen chloride reacts with silver(I) oxide, leading to the formation of the corresponding silver–chlorine complex ([AgCl_2_]^−^) as a Lewis acid (Figure 4), which acts as a necessary catalyst for the formation of interelastomer bonds between chloroprene rubber and hydrogenated acrylonitrile-butadiene rubber. Then, as a result of the detachment of the chlorine atom from the chloroprene rubber, the carbocation is formed, reacting with the double bond derived from HNBR (1,4-*trans*), which leads to the connection of both rubbers (Figure 5), as shown in previous publications [17,18]. The direct use of metal halides (Lewis acids) as a cross-linking agent is not practiced due to the many difficulties encountered during their processing. Such substances are very reactive when rubbers are processing at a higher temperature, and they are characterized by high hygroscopicity and susceptibility to hydrolysis [36]. Therefore, it is recommended to generate aprotic Lewis acids in the elastomer environment (in situ), in the reaction of appropriately selected precursors, including polymeric polyhalide, and to use the acids produced in this way to cross-link elastomers. In the case of the HNBR/CR blend heated in the presence of silver(I) oxide, chloroprene rubber acts as a chloride ion donor, while silver(I) oxide acts as a chloride ion acceptor [17].

During the heating of the HNBR/CR blend, the cross-linking of the chloroprene rubber probably also occurs according to the mechanism shown in Figure 6. In the first stage of CR cross-linking in the presence of silver(I) oxide, the isomerization of chloroprene rubber macromolecules takes place. Then, at an elevated temperature, hydrogen chloride molecules are detached from CR, leading to the formation of a carbocation. The cross-link between the chloroprene rubber macromolecules is formed as a result of the reaction of the carbocation with the double bond of the next CR macromolecule.

### 3.2. Effect of Silver(I) Oxide on Cure Parameters of HNBR/CR Blends

The rheometric properties of HNBR/CR blends were studied to determine the effect of silver(I) oxide as a cross-linking agent of these compounds and to describe the vulcanization process. The vulcametric measurements were carried out at 160 °C, the rheometric curves are presented in Figure 1, and the calculated results are summarized in Table 2.

One of the vulcanization parameters that depends on the type of elastomers used in the blend is the scorch time (t_02_), which defines the safety of elastomers processing at room or higher temperatures. The longer the scorch time is, the greater the processing safety of the rubber and the rubber mix is. All the tested HNBR/CR blends achieved a very short scorch time, because the t_02_ value ranged from 0.24 min to 0.59 min; whereas, the higher the HNBR content is, the longer the scorch time is. Unfortunately, such a short scorch time may cause problems in the processing of the tested blends, especially those with a high content of chloroprene rubber. Therefore, the incorporation of hydrogenated acrylonitrile-butadiene rubber is necessary to improve the processing of HNBR/CR composites. The optimal vulcanization time (t_90_) is an important parameter from an economic and technological point of view. Too long of a vulcanization time is not desirable in the industry, because it means a higher cost for elastomers’ vulcanization and, consequently, a higher cost for the final rubber products. The obtained results indicate that the higher the CR content in HNBR/CR blends is, the shorter the vulcanization time is (t_90_ = 2.94 min for the CR100 mix, and t_90_ = 3.63 min for the HNBR20CR80 blend). In comparison, samples with a higher HNBR content demonstrated a much longer vulcanization time (t_90_ = 13.71 min for the HNBR100 mix, and t_90_ = 13.44 min for the HNBR80CR20 blend). In addition, samples containing a higher content of chloroprene rubber had a higher cure rate index (CRI). The CRI values are very significant. Chloroprene rubber was cross-linked the fastest in the presence of Ag_2_O (CRI = 37.04 min^−1^). Only slightly lower CRI values were determined for blends containing HNBR in the amount of 20–40 phr. In these cases, the cure rate index was equal to 29.76 min^−1^ and 25.84 min^−1^, respectively. In the case of blends with a higher CR content heated in the presence of silver(I) oxide, cross-links between chlorine and silver atoms were more easily formed.

The minimal rheometric torque (T_min_) indicates the viscosity of the uncured HNBR/CR blends. Table 2 shows that the CR100 sample had the highest viscosity, which was confirmed by the largest minimal torque, equal to 1.35 d∙Nm. In the case of the HNBR100 mix, the lowest viscosity and minimal torque (T_m_ = 0.68 d∙Nm) were observed. This is due to the chemical structure of these elastomers, because the content of the chlorine groups in chloroprene rubber indicates stiffness of the polymer chains and, consequently, higher viscosity. Thus, the value of minimal torque decreases with the increasing content of hydrogenated acrylonitrile-butadiene rubber. The susceptibility of CR to crystallization is also of great importance. Chloroprene rubber is one of the elastomers that can crystallize under the influence of low temperature or as a result of orientation brought about by stretching in conditions of moderate or considerable deformation, with the formation of a crystalline phase [37]. This physical behavior of the chloroprene rubber can result in higher viscosity mixes containing a higher amount of this rubber, but problems can arise when processing such elastomeric compositions.

The rheometric torque increment after 15 min of heating (ΔT_15_) is the difference between the rheometric torque determined after a certain heating time (T_15_) and the rheometric minimal torque (T_min_). The torque increment increases when the formation of cross-links begins; therefore, this parameter is related to the cross-linking density of the tested material. The greater the ΔT_15_ value is, the greater the degree of cross-linking of HNBR/CR blends is (Figure 1 and Table 2). The presented results show that the rheometric torque increment increased with the content of chloroprene rubber (ΔT_15_ = 3.54 d∙Nm for the HNBR20CR80 blend, and ΔT_15_ = 1.32 d∙Nm for the HNBR80CR20 blend). In addition, no cross-linking of HNBR was found in the presence of Ag_2_O. From the flat vulcametric curve of the HNBR100 mix, it follows that the hydrogenated acrylonitrile-butadiene rubber was not even slightly cured, because the rheometric torque increment was negligible. These observations indicate the main participation of chloroprene rubber in the cross-linking of HNBR/CR blends in the presence of Ag_2_O; however, even a small amount of CR (20 phr) ensures the cross-linking of the tested compositions, and the kinetics curves show the constant value of torque, which is very important for the technological point of view. In the processing of elastomers, the constant value of the rheometric torque observed on the vulcametric curves is crucial for the vulcanization of thick-walled products requiring extended cross-linking time. In addition, it was noticed that the rheometric torque increment as well as the vulcanization time depend on the proportions of the blends’ components. By increasing the content of the chloroprene rubber in HNBR/CR compounds, an increase in torque and a reduction in vulcanization time were observed. Undeniably, silver(I) oxide can be an effective and fast cross-linking agent for HNBR/CR blends. The cross-linking time of the tested compositions was set at 15 min.

### 3.3. Effect of Silver(I) Oxide on Swelling Behavior of HNBR/CR Vulcanizates

Knowing the densities of the rubbers tested and the solvents used, the equilibrium volume swelling (Q_v_), which is a measure of the degree of cross-linking (α_c_) of the elastomeric compositions, was determined. The equilibrium swelling parameters are summarized in Table 3. According to the obtained results, it can be seen that the tested HNBR/CR vulcanizates tended to swell more in toluene than in n-heptane. In addition, the samples containing only HNBR significantly dissolved in toluene and were in the form of a gel that was difficult to weigh. This fact confirms the impossibility of HNBR cross-linking under the influence of Ag_2_O. Interestingly, as the CR content in vulcanizates increased, the equilibrium swelling value in toluene decreased, even though CR itself also has poor resistance to polar solvents. The highest value of equilibrium volume swelling (Q_V_^T^ = 6.56 mL/mL) was obtained for the HNBR80CR20 sample. The same value (Q_V_^T^ = 6.55 mL/mL) was obtained for the HNBR75CR25 composite, while other samples containing a higher amount of CR yielded lower volume swelling values. The Q_V_^T^ parameter was 4.94 mL/mL for the HNBR20CR80 vulcanizate.

The volume fraction of rubber in the swollen sample (V_r_) is an additional measure of the degree of cross-linking. The higher the V_r_ value is, the more the blend is cured. The data in Table 3 show that the volume fraction of elastomers in the swollen sample was the highest for the CR100 and HNBR20CR80 vulcanizates, for which the V_r_ parameter was equal to 0.17. A much lower value of V_r_, equal to 0.13, was calculated for vulcanizates containing from 75 to 80 phr of HNBR and from 25 to 20 phr of CR, which indicates a significant share of chloroprene rubber in the cross-linking of blends in the presence of Ag_2_O.

In contrast, the limited swelling of the vulcanizates in n-heptane is caused by their different nature (polar elastomers and non-polar solvents). Depending on the solvent used, different effects of the composition of the individual compounds on their properties were observed. When toluene was used, the equilibrium swelling value decreased slightly with increasing CR content in the vulcanizate, and, therefore, its degree of cross-linking slightly increased. Both HNBR and CR showed resistance to aliphatic solvents, although it was greater for HNBR.

The difference in polarity and the different chemical structures of the units in chloroprene rubber and hydrogenated acrylonitrile-butadiene rubber cause the probability of producing a vulcanizate with good functional properties to be limited. In the case of the tested CR/HNBR vulcanizates, increasing the compatibility between the elastomers used is possible through the use of a new cross-linking substance—silver(I) oxide. When HNBR/CR blends were heated in the presence of Ag_2_O, a Lewis acid catalyst was generated in situ. The cross-linking process was confirmed by the values of eluted substances (i.e., the content of the eluted fraction, W_Q_) determined during swelling in toluene (Table 3). Assuming no reactions occurred between CR and HNBR and taking into account the W_Q_ value for vulcanizates containing CR only or HNBR only, as well as the weight fraction of the individual elastomers in HNBR/CR vulcanizates, the theoretical content of the eluted fraction in toluene should be from 0.117 mg/mg to 1.000 mg. The experimentally determined W_Q_ values for HNBR/CR composites were much lower than those calculated. Based on the differences in the theoretically and experimentally determined values of the content of the eluted fraction in toluene, it can be concluded that the heating of HNBR/CR blends in the presence of silver(I) oxide leads to the binding of HNBR to CR (from 14% to 59%). The presented results prove the formation of an interelastomer network in HNBR/CR composites cross-linked with silver(I) oxide.

The results indicated that the degree of cross-linking (α_c_) is higher for compositions containing a higher amount of chloroprene rubber. The degree of cross-linking of the HNBR80CR20 sample was 0.153, but the HNBR20CR80 vulcanizate had an α_c_ value of 0.202. This fact suggests that cross-links are also formed during the curing of chloroprene rubber with Ag_2_O. It is also worth noting that the trend of the swelling degree changes in the same fashion as the trend of the rheometric properties.

### 3.4. Effect of Silver(I) Oxide on Structure of HNBR/CR Composites Observed by Infrared

Attenuated Total Reflectance Fourier-Transform Infrared (ATR-FTIR) spectroscopy was used to explain the structure of HNBR/CR blends before and after their cross-linking in the presence of silver(I) oxide. The infrared spectra confirm that Ag_2_O is an effective curing agent of HNBR/CR composites. The FTIR spectra of the selected HNBR/CR (50/50 by wt.) composite are shown in Figure 2.

The spectra of both uncured and cured HNBR50CR50 composites show two intensive absorption peaks at 2922 cm^−1^,due to the asymmetric stretching vibrations of carbon–hydrogen atoms (ν_as_C-H) in methyl (–CH_3_) and methylene (–CH_2_–) groups, and at 2853 cm^−1^ s, due to the symmetric stretching vibration of –C–H (ν_s_C-H) in the same groups (–CH_3_ and –CH_2_–), which are present in both HNBR and CR chains. Another strong absorption peak appearing in both spectra at 2236 cm^−1^ is attributed to the stretching vibrations of the –C≡N groups in the acrylonitrile units of HNBR. The double band in the range of 1700–1659 cm^−1^ corresponds to double bonds >C=C<, which can be vinyl, trans, and cis, both substituted and unsubstituted. In the range of 1460–1440 cm^−1^, the band of the asymmetric scissoring vibrations of methyl groups (δ_as_CH_3_, 1463 cm^−1^) overlaps with the band of the symmetric scissoring vibrations of methylene groups (δ_s_CH_2_, 1445 cm^−1^), in-plane –C–H deformation in methylene groups. The medium intensive absorption peak at 1300 cm^−1^ is caused by skeletal (γC–C) or deformation (δC–C) vibration between carbon atoms. Both spectra show pronounced peaks at 1347, 1118, 824, and 722 cm^−1^, which are attributed to the symmetric scissoring vibrations of methyl groups (δ_s_CH_3_), the wagging (ωCH_2_) and twisting (τCH_2_) vibrations of methylene groups, out-of-plane –CH=CH_2_ deformations, the vibrations of residual >C=C< bonds, and the stretching vibrations of (νCH_2_)_n_ groups, respectively [38,39,40,41,42].

The absorption peak at 969 cm^−1^ is attributed to the out-of-plane –CH=CH– deformations in trans-butadiene units. It should be emphasized that, in the spectrum of the HNBR50CR50 vulcanizate, the intensity of this peak decreases (Figure 2), which proves that the >C=C<_trans_ bonds present in hydrogenated acrylonitrile-butadiene rubber participate in interelastomer reactions with chloroprene rubber, which occur during the heating of these elastomers in the presence of silver(I) oxide. This absorption change confirms the cross-linking mechanism of HNBR/CR blends with Ag_2_O proposed in Section 3.1. In addition, in the spectrum of the HNBR50CR50 vulcanizate, new absorption peaks appear at 1150, 1050, and 1018 cm^−1^, which are most likely associated with stretching vibrations around nitrogen atoms (νC–N) or deformation vibrations (δN–H). This may indicate the additional transformations of nitrile groups occurring during vulcanization. The strong absorption peaks in the range of 600–650 cm^−1^ probably correspond to the vibrations of C–Cl groups, which confirms the presence of chlorine atoms in both HNBR50CR50 blend and its cross-linking product.

### 3.5. Effect of Silver(I) Oxide on Thermal Analysis of HNBR/CR Vulcanizates

The differential scanning analysis can be the confirmation of interelastomer reactions between hydrogenated acrylonitrile-butadiene and chloroprene rubbers in the presence of silver(I) oxide. Elastomeric blends most often form immiscible systems. The miscibility of several elastomers is designated by the polarity of rubbers and their glass-transition temperatures [43]. In the case of two immiscible rubber compositions, there are two glass-transition temperatures that are unchanged compared to the individual elastomers (unmixed). In the case of partially miscible systems, two glass transitions are observed at temperatures that differ from the respective glass-transition temperatures of the components. In such cases, the T_g_ of the polymer with the lowest glass-transition temperature increases, and the T_g_ of the polymer with the highest glass-transition temperature decreases. In this way, the temperature interval between two glass transitions is shortened. In the case of the perfect miscibility of two polymers, only a single T_g_ is visible [44].

Interelastomer reactions occurring between CR and HNBR can lead to the improved homogeneity and miscibility of the tested compositions. In the DSC curves (Figure 3), it can be observed that there are two glass-transition temperatures, and each of them corresponds to the individual phases of the blend, i.e., CR and HNBR. The glass-transition temperature for pure chloroprene rubber was −41.48 °C, and for pure hydrogenated acrylonitrile-butadiene rubber T_g_ it was −19.75 °C. In the case of blends of both rubbers, changes in the T_g_ values for individual elastomers were observed. For a blend containing 50 phr of CR and 50 phr of HNBR, the glass-transition temperature for chloroprene rubber increased to −40.41 °C, and for hydrogenated acrylonitrile-butadiene rubber T_g_ it decreased to −23.65 °C (Table 4). These changes may indicate the presence of interactions and increased compatibility between HNBR and CR. The DSC result was surprising for the HNBR20CR80 composition, for which only one glass-transition temperature (T_g_ = −30.40 °C) was observed. One glass transition may prove the complete miscibility of HNBR with CR and the accompanying interelastomer reactions, which are formed under the influence of silver(I) oxide used to cross-link such blends.

The DSC thermogram is the result of all reactions occurring in a selected temperature range. It should be remembered that during the heating of the elastomeric composite, many reactions competitively occur; thus, it is very difficult to measure the enthalpy for a specific reaction. On the DSC curves of all HNBR/CR blends and the CR100 composite, a distinct endothermic peak at approx. 45 °C was found that corresponds to the crystalline phase of chloroprene rubber. This phenomenon is a typical behavior of CR, which is able to crystallize during storage or stretching and becomes amorphous during heating [45]. In addition, on the DSC curve of the HNBR100 mix, a wide and significant endothermic peak at approx. 0 °C was observed. Most likely, this peak corresponds to the crystallization of hydrogenated acrylonitrile-butadiene rubber.

The cross-linking of HNBR/CR blends took place in the temperature range from 119 °C to 232 °C. The enthalpy determined during curing can be correlated with the degree of cross-linking. The higher the enthalpy value is, the more efficacious the cross-linking of blends is, and, thus, the greater the α_c_ value of the investigated composites is. In the case of HNBR/CR blends heated in the presence of silver(I) oxide, the exothermic peaks were broad and low with comparable values of enthalpy (ΔH = 15.50–17.42 J/g). The cross-linking progress is mainly indicated by the type of rubber and cross-linking substance used. In these studies, only low intensities of exothermic peaks were observed. This probably indicates a lower efficiency of silver(I) oxide as a cross-linking agent for HNBR/CR composites. For comparison, in the case of cross-linking of blends containing chloroprene and butadiene rubbers with nano-iron(III) oxide, a very intense exothermic peak was visible [46]. In the case of pure chloroprene rubber cured with silver(I) oxide in a higher amount, 5 phr, an intense exothermic peak was observed [11]. It follows that a smaller amount of silver(I) oxide resulted in a lower efficiency of the cross-linking process.

The cross-linking temperature range for the investigated HNBR/CR blends was from 127 °C to 201 °C. Such a low onset temperature of cross-linking most likely indicates the pre-curing process, to which both HNBR and CR are particularly susceptible. Silver(I) oxide contributes in particular to the pre-curing of HNBR and CR blends, which was easy to observe during practical vulcanization on hydraulic presses. The tested blends were very quickly scorched, which can also be confirmed by the very short scorch time (Table 2). The obtained results from the DSC thermogram correlate with the obtained results from the vulcametric curve and equilibrium swelling. As previously mentioned (Table 2 and Table 3), the HNBR80CR20 and CR100 compounds were characterized by the highest degree of cross-linking. Unexpectedly, on the DSC curve of HNBR, a significant exothermic peak was recorded in the range of 190 °C–210 °C, i.e., in the range where no exothermic peaks were observed for the tested HNBR/CR blends. This indicates an additional chemical reaction taking place in HNBR under the influence of very high temperatures. According to the literature data, HNBR under such conditions is already subject to destruction, which mainly concerns cyan units [30]. Thermal degradation begins with oxygen attack at the α-methylene groups at >C=C< bonds. The destruction of hydrogenated acrylonitrile-butadiene rubber is dominated by the involvement of -CN units in the structuring processes, including the generation of new intra- and/or intermolecular bonds.

### 3.6. Surface Morphology of HNBR/CR/Ag_2_O Vulcanizates

The SEM analysis provided additional information on the miscibility of the components in HNBR/CR composites. In addition, the SEM analysis allows for the estimation of the dispersion of silver(I) oxide in the investigated blends. The surface morphologies of tested vulcanizates at a 3k magnification are shown in Figure 4.

The SEM image of the HNBR100 sample confirms a proper dispersion of silver(I) oxide in the HNBR matrix and shows a unitary elastomer surface with a lighter discoloration corresponding to small Ag_2_O aggregates unevenly distributed in the rubber (Figure 4a). In the SEM image of the HNBR80CR20 vulcanizate, only single small Ag_2_O aggregates were visible. This demonstrates the droplet dispersion of chloroprene rubber in the HNBR matrix (Figure 4b). In Figure 4c, minor surface cracks of the HNBR75CR25 vulcanizate and clear CR microphases droplets dispersed in the HNBR matrix were depicted. A more homogeneous structure of the HNBR and CR compositions is shown in Figure 4d, although the limited miscibility of both elastomers should also be pointed out here. Probably, the cross-linking of blends characterized by partial miscibility occurs at the phase separation boundary instead of throughout the elastomeric mass, as in the case of totally homogeneous blends. However, the SEM image of the HNBR50CR50 vulcanizate confirms better miscibility of this sample, except for one large rubber agglomerate located on the left side of the image (Figure 4e). It can be seen that the CR phase became the dominant phase in the HNBR50CR50 vulcanizate. Numerous clusters of Ag_2_O are visible here. The SEM image of the HNBR40CR60 vulcanizate confirms good miscibility of HNBR and CR phases but shows the insufficiently good dispersion of silver(I) oxide in the matrix of both elastomers (Figure 4f). In Figure 4g, which presents the SEM image of the HNBR20CR80 vulcanizate, a clear agglomerate of Ag_2_O in the form of a white line is shown. There are no visible rubber microphases in the investigated sample. This indicates the proper distribution of HNBR in the CR matrix, i.e., the high degree of homogeneity of this composition. In contrast, in the CR100 vulcanizate, apart from two aggregates, a uniform dispersion of Ag_2_O in the elastomer matrix was observed (Figure 4h).

For the HNBR50CR50 vulcanizate, an additional composition study using the SEM-EDS method was performed. Figure 5 shows the spectrum of elements present in the tested sample. The obtained spectrum confirms the presence of silver, chlorine, carbon, and oxygen atoms in the HNBR50CR50 vulcanizate. An intense signal at 2.622 keV came from both chlorine and silver atoms. An additional signal indicating silver atoms was recorded at 2.984 keV. The spectrum also shows a peak from aluminum atoms, which came from the microscope column or table used in the study. The low-intensity signal at 2.307 keV came from sulfur, which was used to modify the chloroprene rubber during its polymerization. The distribution of elements in selected micro-areas indicates a uniform dispersion of the indicated elements, including silver atoms (Figure 6). Slightly brighter areas, visible in Figure 6f, indicate the presence of small aggregates of silver(I) oxide in the tested HNBR50CR50 vulcanizate, which is also confirmed by the SEM image made for this material (Figure 4e).

### 3.7. Effect of Silver(I) Oxide on Mechanical Properties of HNBR/CR Vulcanizates

The mechanical properties of the HNBR, CR, and HNBR/CR composites were determined (Table 5). From the obtained results, it was found that the tensile properties of the tested vulcanizates depend on the proportions of their components. The first considered parameter is stress at a relative elongation of 100%, 200%, or 300% (S_e100_, S_e200_, and S_e300_, respectively). Stress at elongation of 100% is particularly important because it determines the stiffness of tested samples. Unexpectedly, the obtained stress values for HNBR/CR vulcanizates were higher than for the HNBR sample or CR vulcanizate. For example, the S_e100_ parameter was 1.25 MPa for the HNBR20CR80 vulcanizate, but this property was only 0.48 MPa and 0.89 MPa for the HNBR100 and CR100 samples, respectively. This points to the higher stiffness of cured blends rather than the stiffness of chloroprene rubber, which is known as a very high-viscosity rigid rubber. Similar dependences were observed in the case of stresses at 200% or 300% elongation. All HNBR/CR blends cross-linked with Ag_2_O had higher S_e200_ and S_e300_ parameters than HNBR or CR. Of course, it should be remembered that HNBR was not cross-linked with silver(I) oxide and that its mechanical parameters were set as the reference points for the properties of other vulcanizates.

The most important parameter is tensile strength (TS_b_) as the maximum stress when the specimen breaks during elongation [47]. The results of the study showed that in the case of TS_b_, the obtained values proportionally increased with the increasing content of CR. The highest mechanical strength of HNBR/CR composites was obtained by the HNBR20CR80 sample (TS_b_ = 7.63 MPa). The tensile strength of CR100 vulcanizates was 17% higher and amounted to 8.93 MPa. The reason for this behavior of HNBR/CR vulcanizates may be related to the fact that interelastomer reactions occur between HNBR and CR heated in the presence of silver(I) oxide, and the resulting better miscibility of both elastomers was demonstrated by the SEM images (Figure 4). When the CR amount increased, both elastomers were more partially compatible, so the mechanical properties of the material were better and reached the maximal value at a CR content of about 80% (Table 5). The obtained results confirm high mechanical strength, considering that the tested vulcanizates did not contain any filler, and no accelerator was used for cross-linking with silver(I) oxide. For comparison, the tensile strengths of HNBR/CR blends cured with sulfur, zinc, and magnesium oxides, in the presence of tetramethylthiuram disulfide and 2-mercaptobenzothiazole as accelerators, were only in the range of 5–8 MPa [48].

HNBR/CR/Ag_2_O vulcanizates were characterized by different elongation at break (E_b_). The highest elongation at break (792%) was shown by the HNBR80CR20 sample (Table 5). The elongation at break increased with the higher content of HNBR; however, for the HNBR100 sample, it reached 77%. The higher E_b_ values recorded for vulcanizates with a predominant amount of HNBR testify to the greater elasticity of such samples. This is of great practical importance because in many applications rubber materials should be highly flexible. These observations correlate with the hardness results of the tested vulcanizates. The higher amount of chloroprene rubber in HNBR/CR vulcanizates increased their hardness, compared to this parameter in the HNBR100 sample (50 °ShA). However, above a content of 50 phr of CR, the measured hardness of the cross-linked blends was comparable to that of the CR100 vulcanizate (approx. 60 °ShA). Due to its chloride groups, chloroprene rubber is a very stiff and hard rubber, so its addition has a major effect on the hardness of the elastomeric blend. However, all vulcanizates subjected to thermo-oxidative aging showed reduced hardness. The higher the CR content in HNBR/CR compositions was, the greater the observed percentage decrease in hardness was. For the CR/Ag_2_O vulcanizate, the reduction in hardness was 40%.

Tear resistance (T_s_) is very important in many rubber applications. Usually, this parameter is reported as a force required to pull a rubber test piece apart using a tensile testing machine under controlled conditions [49]. In this study, a special cut was applied to the specimen. The tear resistance results show that, initially, a small addition of CR increased the tear resistance of the vulcanizates. However, with increasing chloroprene rubber content, the force required to tear the tested products gradually decreased (Table 5). From a content of 40 phr of CR upward, the determined tearing stresses were lower than the value recorded for pure CR. Considering that mechanical properties are largely dependent on the cross-link density, it can be concluded that the addition of more easily cross-linked CR increases the cross-link density of the composition and, until a certain point, also the tear resistance. Above this value, an increasing degree of cross-linking results in a worsening of the mechanical resistance of the vulcanizate. The HNBR80CR20 sample achieved the highest tear strength (18.2 N/mm), while the HNBR20CR80 vulcanizate had the lowest (6.1 N/mm).

### 3.8. Effect of Silver(I) Oxide on Resistance to Thermo-Oxidative Aging of HNBR/CR Vulcanizates

Premature and undesirable aging of rubber materials is a common disadvantage and a major limitation in many applications requiring long rubber life. For this reason, it is extremely important to produce elastomeric materials that show high resistance to factors accelerating the aging of such products. Therefore, the resistance of the produced HNBR/CR/Ag_2_O vulcanizates to thermo-oxidative aging was tested. The influence of thermo-oxidative aging was primarily studied using stress at elongation of 100%, tensile strength, and elongation at break (Table 5). It was concluded that subjecting HNBR/CR samples to the thermo-oxidative aging process resulted in a reduction in the stress values at a specific relative elongation (S_e_*). The mechanical properties of the HNBR100 sample subjected to an elevated temperature remained unchanged, but it was already emphasized that this elastomer was not cross-linked with silver(I) oxide, so the assessment of its properties is only informative. In the case of HNBR/CR blends cross-linked with Ag_2_O and subjected to aging factors, a decrease in the S_e100_ parameter by approx. 50% was found, which indicates significantly lower stiffness of the tested samples. Only in the case of the CR100 vulcanizate, the stress at elongation of 100% remained unchanged after thermo-oxidative aging (0.89 and 0.90, respectively) (Table 5 and Figure 7).

The HNBR100, HNBR40CR60, and HNBR20CR80 vulcanizates showed little change in tensile strength under thermo-oxidative aging. The cross-linked HNBR75CR25 compound showed the greatest decrease in tensile strength, equal to almost half of the value determined before aging (TS_b_ = 4.43 MPa, and TS_b_* = 2.26 MPa). The tensile strength for the HNBR20CR80 vulcanizate increased from 7.63 MPa to 7.83 Mpa, indicating the high resistance of this product to thermo-oxidative aging. Surprisingly, for the CR100 vulcanizate, a clear improvement in tensile properties was observed after aging. The TS_b_ value of CR cured with Ag_2_O was 8.93 MPa, and after exposure to high temperature for 7 days the CR100 vulcanizate showed a 50% increase in tensile strength (TS_b_* = 13.3 MPa). This is probably due to the use of silver(I) oxide as a cross-linking agent and the extra curing phenomenon (Table 5 and Figure 8).

The elongation at break (E_b_*) of all vulcanizates subjected to thermo-oxidative degradation increased, though only slightly for the HNBR50CR50 vulcanizate (E_b_ = 616%, and E_b_* = 642%). The greatest increase in elasticity after aging factors was evident for the HNBR80CR20 vulcanizate, which had a higher deformation capacity (E_b_ = 792%, and E_b_* = 1151%) than the other vulcanizates (Table 5 and Figure 9).

Both HNBR and CR are elastomers categorized as aging-resistant materials. For this reason, their vulcanizates could also be expected to show similar properties. The calculated aging factor (AF) values for most vulcanized materials oscillate around 1, which confirms high thermo-oxidative resistance (Table 5 and Figure 10). Only the HNBR75CR25 vulcanizate, with an AF value of 0.54, proved to be susceptible to aging processes, resulting in a significant deterioration of its strength properties. The CR100 vulcanizate showed the greatest resistance to thermo-oxidative degradation processes. In this case, the calculated aging factor was as high as 2.17. However, it can only be speculated that this cross-linking with Ag_2_O allowed the mechanical properties of pure CR to double under aging process. Among the cured blends, the highest AF parameter (1.13) was achieved by the HNBR20CR80 and HNBR40CR60 vulcanizates.

## 4. Conclusions

In this paper, the development of unconventional cross-linked blends of hydrogenated acrylonitrile-butadiene and chloroprene rubbers (HNBR/CR) was described. It was found that HNBR/CR blends and the CR compound were effectively cross-linked with silver(I) oxide (Ag_2_O). The degree of cross-linking of HNBR/CR/Ag_2_O compositions depended on the weight proportion of both elastomers, and it was greater with more content of chloroprene rubber in tested blends.

Cross-linking of HNBR/CR blends with Ag_2_O leads to the binding of HNBR with CR, and the degree of binding was in the range of 14–59%. The presented results prove the formation of interelastomer reactions between hydrogenated acrylonitrile-butadiene rubber and chloroprene rubber, which leads to the improved homogeneity and miscibility of the tested compositions. In the DSC curves, the temperature interval between the glass-transition temperature of HNBR and the glass-transition temperature of CR was shortened. However, in the case of the HNBR/CR/Ag_2_O (20/80/2.5 by wt.) blend, only a single T_g_ (−30.40 °C) was visible. These changes indicate the presence of interactions and increased compatibility between HNBR and CR under the influence of silver(I) oxide. The SEM analysis provided additional information on the miscibility of the components in HNBR/CR/Ag_2_O composites.

HNBR/CR blends cured with Ag_2_O showed good mechanical properties. The tensile strength of the obtained vulcanizates proportionally increased with the increasing content of CR. The highest mechanical strength was obtained in the case of the HNBR20CR80 vulcanizate (TS_b_ = 7.63 MPa), but the tensile strength of the CR100 vulcanizate was 17% higher and amounted to 8.93 MPa. The higher elongation at break recorded for vulcanizates with a predominant amount of HNBR testifies to the greater elasticity of such samples. The hardness increased with the increasing amount of CR in HNBR/CR vulcanizates, while the tear strength showed an inverse relationship, i.e., it increased with the increasing content of HNBR in the tested materials.

The obtained new, unconventional materials were characterized by significant resistance to thermo-oxidative factors, which was confirmed by the high aging factor. Therefore, this type of HNBR/CR/Ag_2_O vulcanizates can be applied in the production of technical rubber products with high thermal stability as well as high environmental resistance.

## Data Availability

Not applicable.

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
