# Peer review of "Interelastomer Reactions Occurring during the Cross-Linking of Hydrogenated Acrylonitrile-Butadiene (HNBR) and Chloroprene (CR) Rubbers Blends in the Presence of Silver(I) Oxide (Ag_2_O) and Mechanical Properties of Cured Products"

_materials, 2023, doi:10.3390/ma16134573_

Round 1
Reviewer 1 Report
This paper presents an experimental investigation on HNBR/CR blends and proposes silver oxide as a crosslinking agent. The paper is generally well presented and discussed, but some points need attention:
p.2 : change « specialist » by « speciality ».
p.3: remove the line change for the 3rd and after the 4th bullet point. Several other places: revise all document.
p.4: change « (PET) foils were » by “PET foil was”.
p.6: it is not clear how the 2.5 phr value was selected for the silver oxide. It would be more interesting to produce a series with different content to determine this “optimum” content which should also be the objective of this paper to be more complete… Maybe for another study in the near future ?
Figure 1: difficult to differentiate the curve without color. Use numbers ? You can cut the time scale at 15 min to better see the initial time changes.
Figure 2: difficult to differentiate the curve without color. Use numbers ? Maybe shift the curve for better readability ?
p.14: change “DCS” to “DSC”.
p.15: change “popper” by “proper”.
Figure 3. Maybe perform an Ag mapping to get more information on its distribution…
Table 4. Hardness after aging is missing ?
p.18: change “MPA” to “MPa” (two places).
Figure 5: change “Strenght” to “Strength”.
Conclusion: revise for subs/superscripts.
Right align the equation numbers.
The study can be complemented with TGA and FTIR analyses to get more information on the links produced and their stability.
ok.
Author Response
Thank you for your thorough review of my article. All comments are very important to me. My review notes and corrections to the article are attached.

Reviewer 2 Report
Following are some of the observations that should be considered to improve the quality of content.
· The abstract needs thorough revision. It should be concise and reflect the essence of the research. (e.g. grammatical mistakes in line 4, Sentences are too lengthy to comprehend, Line 11 the term “ research” does not fit here appropriately. Lines 11-17 needs revision.
· The introduction sections need critical revisions and re-writing. Grammatical issues and repetition of sentences need to be avoided. A few sections also lack connectivity. At the end of the introduction sections, the research gap should be followed by possible application areas of the proposed material.
· In the material section (2.2), Unnecessary bullet points are used. While there are lots of mistakes for commas, semicolons, spaces, and line breaks. The author needs to re-write a material section in a uniform format.
· Research methods lack sequence. There is no proper description of the blends' preparation method and composition. The authors need to describe the step-by-step preparation method for the blends.
· Section 3.3 can be presented at the start of the result and discussion section followed by 3.1 and 3.2. The author needs to co-relate the curing, crosslinking, and swelling ratio to eliminate the redundant details.
· In most of the studies, a distinct and sharp exothermic peak is observed corresponding to the crosslinking of chloroprene and its blends. A similar trend has been observed in the author’s previous publication. But this study (Figure 2) shows a different DSC curve for chloroprene and blends without distinct exothermic peaks (DOI: 10.3390/polym13060853). The authors need to add more explanation regarding this.
· In section 3.5, Different terminologies have been used for description. i.e. SEM image, SEM photo, and SEM picture. Single terminology should be used throughout the manuscript. Furthermore, SEM images are not clear. The author also needs to revise the images and label the description like agglomerates or clusters etc.
· In Figure 5 and Figure 6, Tensile strength and elongation at break both have shown an increment for HNBR20CR80 which does not seem to be in line with the provided explanation.
The conclusion can be made more concise by eliminating the explanations and highlighting only the clear-cut outcomes of the research.
Extensive editing of the English language is required.
Author Response

(The authors gave the same response as above.)

Round 2
Reviewer 2 Report
Accept